# COLLU-BENCH: A BENCHMARK FOR PREDICTING LANGUAGE MODEL HALLUCINATIONS IN CODE

## ABSTRACT

Despite their success, large language models (LLMs) face the critical challenge of hallucinations, generating plausible but incorrect content. While much research has focused on hallucinations in multiple modalities including images and natural language text, less attention has been given to hallucinations in source code, which leads to incorrect and vulnerable code that causes significant financial loss. To pave the way for research in LLMs' hallucinations in code, we introduce *Collu-Bench*, a benchmark for predicting code hallucinations of LLMs across code generation (CG) and automated program repair (APR) tasks. Collu-Bench includes 13,234 code hallucination instances collected from five datasets and 11 diverse LLMs, ranging from open-source models to commercial ones. To better understand and predict code hallucinations, Collu-Bench provides detailed features such as the per-step log probabilities of LLMs' output, token types, and the execution feedback of LLMs' generated code for in-depth analysis. In addition, we conduct experiments to predict hallucination on Collu-Bench, using both traditional machine learning techniques and neural networks, which achieves 22.03 – 33.15% accuracy. Our experiments draw insightful findings of code hallucination patterns, reveal the challenge of accurately localizing LLMs' hallucinations, and highlight the need for more sophisticated techniques.

## 1 INTRODUCTION

Despite the great potential and impressive success of LLMs (Touvron et al., 2023; Brown et al., 2020; Li et al., 2022a; OpenAI, 2024), a known issue of LLMs is *hallucination*, a phenomenon where the model generates fluent and plausible-sounding but unfaithful or fabricated content (Ji et al., 2023). The hallucination issue poses a significant risk when deploying LLMs in real-world applications that require precise information (Puchert et al., 2023). Due to this importance, researchers have developed benchmarks such as TruthfulQA (Lin et al., 2022), FELM (chen et al., 2023), and HaluEval (Li et al., 2023b) to understand and predict hallucinations of LLMs. Additionally, researchers are actively exploring methods to mitigate hallucinations (Liu et al., 2024b; Elaraby et al., 2023; Dhuliawala et al., 2023; Yan et al., 2024).

Another domain where LLMs have been widely applied is source code. LLMs are used in many code-related applications, such as code generation (Wang et al., 2023; Li et al., 2023a; Guo et al., 2024; Lozhkov et al., 2024; Rozière et al., 2024), automated program repair (Hossain et al., 2024a; Ruiz et al., 2024; Silva et al., 2023; Jimenez et al., 2024; Hossain et al., 2024b; Jiang et al., 2023; Xia et al., 2023), and software engineering agents (OpenAI, 2024; Yang et al., 2024; Zhang et al., 2024). Unfortunately, in the code domain, LLMs also face the risk of hallucination, such as generating misused Application Programming Interfaces (APIs), insufficient error handlers, or even vulnerable code. Such hallucinations can cause the breakage of code bases, the shutdown of services, exploitation of vulnerabilities, and eventually lead to huge financial costs [1].

Although important, hallucination in code is much less explored compared to that in natural language text and images. Some existing work explores the API misuse issue (Zhong & Wang, 2024), and there are a few benchmarks for hallucination in code generation tasks, categorizing code hallucination into different types (Liu et al., 2024a; Tian et al., 2024). Nevertheless, they only recognize the

---

[1] https://cybersecurityventures.com/cybercrime-bytes-10-hot-security-certs-public-safety-hacked-intrusions-shield/

existence of hallucinations without detecting, predicting, or localizing the hallucinated part. These benchmarks lack analysis of hallucination in code in a finer granularity. Instead of evaluating the entire code, we want to identify the specific token where the hallucination occurs and analyze the characteristics of code hallucinations. A dataset with such information can facilitate a deeper understanding of code hallucination and make it possible to develop targeted and efficient techniques to mitigate the code hallucination issue.

To fill this gap, we introduce *Collu-Bench*, a benchmark to evaluate and analyze code hallucinations in LLMs. Collu-Bench targets two important LLM applications in coding: code generation (CG) and automated program repair (APR). We design an automated pipeline and build the benchmark on five datasets using 11 LLMs with various structures and sizes. In total, Collu-Bench includes 13,234 code hallucination instances. To facilitate the understanding of where the LLM makes mistakes, Collu-Bench includes detailed signals such as per-step log probabilities (prob.), token types, and execution feedback. Such signals reveal the patterns of LLMs' hallucinations in code and benefit the development of techniques to predict and localize hallucinations efficiently in advance.

We conduct a preliminary investigation of localizing code hallucinations on Collu-Bench by training different models, ranging from traditional machine learning (ML) approaches (random forest, etc.) to neural network (NN) models (LSTM, etc.). The goal is to predict the hallucination in the code generated by LLMs in an *efficient* and *lightweight* way, by observing the behavior pattern (such as log probs. of tokens during generation) of the targeting LLMs. *Such prediction aims to help the targeting LLMs reflect in time and thus produce more accurate code, instead of replacing the LLMs*. We set up the code hallucination localization task in two ways: per-token prediction, and per-sample prediction. We further set up the data split in three ways: All-in-one (building a universal predictor for all LLMs and on all data domains), One-per-dataset (building a predictor on each data domain), and One-per-LLM (building a predictor for each LLM). Our comprehensive experiments draw insightful findings in code hallucination of LLMs.

The main contributions of this paper are as follows:

- We build Collu-Bench, a benchmark with 13,234 code hallucination instances produced by 11 LLMs on five datasets. Collu-Bench includes detailed information such as per-step log prob., token types, and execution feedback, which are useful signals for developing code hallucination localizing and predicting techniques.
  - We propose an automated pipeline, by sampling equivalent code and program normalization, to collect more accurate hallucination token locations during the construction of Collu-Bench.
- We conduct preliminary yet comprehensive studies of code hallucination localization using Collu-Bench, and the key findings are as follows:
  - LLMs are less confident when hallucinating, as the hallucinated tokens have lower prob. and hallucinated generation steps have higher entropy (Section 4.1).
  - LLMs are more likely to hallucinate when generating certain types of tokens such as `Keyword`, `Identifier`, and `Type Identifier` (Section 4.1).
  - When conducting per-token prediction of hallucination token, random forest produces the highest overall accuracy of 33.09%. When conducting per-sample prediction of hallucination location, LSTM produces the highest overall accuracy of 33.15% (Sections 5.1 and 5.2).
  - Under "One-per-dataset" and "One-per-LLM" settings, per-token and per-sample predictions show different patterns and complement each other (Sections 5.1 and 5.2).
- Our results with overall accuracy ranging from 22.03% to 33.15%, show that code hallucination prediction and localization is still a challenging task having large space to improve.

**Availability:** The benchmark is available at `https://zenodo.org/records/13877115`.

## 2 RELATED WORK

### 2.1 TEXT AND IMAGES HALLUCINATION BENCHMARKS

Hallucination in natural language generation (NLG) refers to the phenomenon where models generate text that is fluent but factually incorrect or inconsistent with the input data. Several benchmarks and studies have been proposed to address this issue. HaluEval is a large-scale hallucination evaluation benchmark designed to assess the performance of large language models (LLMs) in gener-

ating factually accurate text (Li et al., 2023b). It provides a comprehensive collection of generated and human-annotated hallucinated samples. FELM introduces a benchmark designed to evaluate the factuality of text generated by LLMs across diverse domains, including math, reasoning, and world knowledge (chen et al., 2023). HaDes is a token-level reference-free hallucination detection benchmark, providing a fine-grained analysis of model performance without relying on ground truth references (Liu et al., 2022). Additionally, RARR uses language models themselves to research and revise the factual consistency of their outputs (Gao et al., 2023).

In the multi-modal tasks. MHaluBench is a comprehensive benchmark for evaluating hallucinations in multi-modal settings (Chen et al., 2024), which incorparates a wider range of hallucination categories and tasks, such as image-to-text and text-to-image generation. MHaluBench offers fine-grained annotations that help identify hallucinations at a detailed level, and facilitates a deeper understanding of hallucination in MLLMs and provides a robust foundation for improving model reliability in practical applications.

## 2.2 Code Hallucination Benchmarks

HalluCode (Liu et al., 2024a) explores hallucinations in the context of code generation. It introduces a comprehensive taxonomy of hallucinations specific to LLM-powered code generation, categorizing them into five primary types. The authors conducted a thematic analysis of LLM-generated code to classify hallucinations based on deviations from user intent, internal inconsistencies, and misalignment with factual knowledge. The benchmark evaluates LLMs' ability to recognize and mitigate hallucinations, revealing that current models face significant challenges.

CodeHalu (Tian et al., 2024) focuses on investigating code hallucinations through execution-based verification. The authors categorize code hallucinations into four main types: mapping, naming, resource, and logic hallucinations, each of which highlights unique challenges in code generation. CodeHalu presents a dynamic detection algorithm to detect and quantify hallucinations and introduces the CodeHaluEval benchmark, which includes a large set of samples to evaluate LLM performance in code generation.

Collu-Bench differs from both HalluCode and CodeHalu in two key aspects. First, Collu-Bench focuses on identifying *where* the hallucination occurs by pinpointing the exact token at which the model first deviates from the expected output. Second, Collu-Bench provides additional signals, such as the types of generated tokens, helping researchers better understand the underlying patterns of code hallucinations.

## 3 Benchmark Construction

In this section, we describe the collection process of Collu-Bench. We first describe our automated pipeline of handling program equivalency and identifier viability, which helps in collecting accurate hallucination token locations in Collu-Bench (Section 3.1). Then we introduce the selected datasets and LLMs in Section 3.2. Section 3.3 shows the process of using LLMs to generate outputs and collect the hallucination token index automatically. Lastly, in Section 3.4, we explain the additional signals Collu-Bench includes, that could help localize hallucination tokens in LLM-generated code.

### 3.1 Handling Code Equivalence and Variation

A standard approach for localizing the hallucinated token is to compare the generated solution with the canonical solution. However, simply comparing the canonical solution and the generated code can lead to many false positives, since the LLM may follow an alternative way to solve the task (Li et al., 2022b; Austin et al., 2021; Chen et al., 2021a;b). For instance, the task of sorting a list of integers can be implemented with many different sorting algorithms. Even semantically equivalent solutions may have a range of syntactic variations, e.g., naming variables differently, using a `for` loop instead of a `while` loop, etc.

Existing hallucination benchmarks in natural language or vision domains although face similar challenges of diversity in text, they can manually annotate the hallucinations in text or images. Compared to text or images, hallucination in code is much more complex and harder to label, as it requires domain expertise. To build a large benchmark of hallucination in code, we propose a pipeline of

collecting diverse correct solutions and normalizing programs to automate the calculation of hallucination location in LLM-generated code.

**i). Diverse Canonical Solution Collection:** For each problem in the dataset, besides the official canonical solutions, we enhance the diversity of canonical solutions by using LLMs to sample more.

For the CG task, due to the simplicity of coding problems in HumanEval and MBPP, there could be lots of different algorithms solving the problems correctly. To cover the equivalent canonical solutions as much as possible, we let each LLM (DeepSeek-Coder-1.3b/6.7b, StarCoder2-3b/7b/15b, CodeLlama-7b/13b, Llama3-8b, and GPT-4o-mini) sample 100 programs per problem, using a temperature of 0.8. These sampled programs are run against EvalPlus for evaluation of correctness, and those that pass all the test cases are considered equivalent canonical solutions.

For the APR task, we conduct the same sampling process (i.e., each LLM sample 100 outputs per repair problem and run against test cases) for the HumanEval-Java dataset to collect canonical solutions, given its simplicity. For Defects4J and SWE-Bench, since (1) the program repair problems in these two datasets are much more complex and thus are less likely to have many diverse equivalents, and (2) their execution of test cases are computationally expensive, we do not conduct sampling and only consider the developer fix provided in the datasets, as well as LLM-generated fixes using greedy decoding that pass all the test cases, as the canonical solutions.

**ii). Program Normalization:** Collecting diverse canonical solutions is effective in covering correct programs implemented with different algorithms or logic. However, it cannot account for the limitless variants of identifier names that can be used within the same program. For example, "`for x, y in zip(tup1, tup2)`" and "`for a, b in zip(tup1, tup2)`" are logically equivalent but differ textually due to the use of different identifier names. Thus, we conduct program normalization to replace all the user-defined identifiers with normalized names so that different choices of identifier names will not be considered hallucinations.

We use tree-sitter (Brunsfeld et al., 2024), a static parser, to parse the generated code into AST, and walk through the AST to collect all the user-defined identifiers. Details can be found in Appendix A.1. After collecting a set of unique user-defined identifiers from a program generated by an LLM (e.g., collecting the identifiers {`a, b`} from the code snippet "`for a, b in zip(tup1, tup2)`", which is a "for statement" in Python), we rename these identifiers sequentially as `v1`, `v2`, and so on, to normalize the program. For instance, `a` is replaced by `v1` and `b` is replaced by `v2`, thus code snippet "`for a, b in zip(tup1, tup2)`" is normalized into "`for v1, v2 in zip(tup1, tup2)`". During this step, the logically equivalent programs with different identifier names will be normalized into the same program.

## 3.2 DATASETS AND LLMS

We target two code-related tasks in Collu-Bench: code generation (CG) and automated program repair (APR). In total, we select five datasets to build the benchmark.

**Code generation (CG):** Code generation is the task of automatically producing code from natural language descriptions. It plays a crucial role in software development by improving productivity and enabling non-programmers to create code through high-level specifications. It is widely used to evaluate the coding capability of LLMs. We use the following CG datasets to build Collu-Bench:

- **MBPP** (Austin et al., 2021): MBPP is a code generation benchmark comprised of hand-written problems solvable by entry-level Python programmers. We use the sanitized version from EvalPlus (Liu et al., 2023) which contains 343 problems.
- **HumanEval** (Chen et al., 2021a): The HumanEval benchmark contains 164 hand-written Python programming problems with function signatures, docstrings, and unit tests.

**Automated Program Repair (APR):** Automated program repair is the process of automatically fixing bugs in software programs, which can significantly reduce the time and effort required for manual debugging and repair. We use the following APR datasets to build Collu-Bench:

- **HumanEval-Java** (Jiang et al., 2023): A benchmark for APR in Java that is transformed from HumanEval to overcome the data leakage threat of Defects4J. It contains 164 injected bugs using 27 diverse mutation rules.

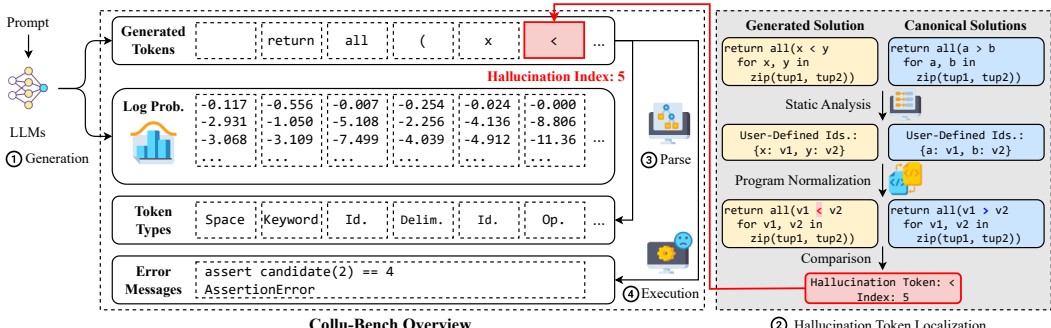

Figure 1: Overview of the benchmark construction

- **Defects4J** (Just et al., 2014): A widely used benchmark for APR in Java. It contains bug fixes from popular open-source Java projects. We use the 235 single-hunk bugs (where the buggy code and corresponding fixed code are within a continuous code chunk) in the Defects4J as a simpler starting point following existing APR techniques (Jiang et al., 2023; Hossain et al., 2024a).

- **SWE-bench** (Jimenez et al., 2024): A recent dataset for project-level program repair in Python, collected from the merged pull requests of popular Python libraries on GitHub. Similarly, we use a subset of 792 single-hunk bugs.

We include outputs of 11 LLMs of five series in Collu-Bench, including open-source ones and commercial ones with different sizes in each category to cater to different researchers' interests. This selection covers open-source code-specialized (DeepSeekCoder, StarCoder2, and CodeLlama) and general (Llama3) models with sizes smaller than 34B and one of the state-of-the-art commercial models (GPT-4o-mini). Additional details of the selected LLMs such as their sizes and release dates are provided in Appendix A.3.

### 3.3 GENERATION AND AUTOMATED HALLUCINATION LOCALIZATION

Figure 1 illustrates the generation step that collects the LLMs' outputs for given coding or repairing problems, and the hallucination token localization step which automatically calculates the index of the first generated hallucination token.

**Code Generation:** For each sample in the datasets (HumanEval, MBPP, etc.), we let each LLM generate one solution code using few-shot prompting (Brown et al., 2020) and greedy decoding. Details and examples of the prompt we used to collect LLMs generated code are provided in Appendix A.2.

**Localization of Hallucinated Tokens:** This step collects the hallucination token indices from the incorrect generate code by normalizing it and comparing it with the large, diverse set of canonical code (Section 3.1), as these will be the targets of Collu-Bench. Specifically, we compare the LLM-generated program with canonical solutions to decide the hallucination location. We normalize the generated code and compare it with each normalized solution one by one. Non-indentation white space in Python programs and all white space in Java programs are ignored during the comparison as they do not affect functionality. The first different character is mapped back to the original generated code before normalization to locate the token where this mismatched character is from.

For instance, in the example shown in Figure 1, the normalized LLM-generated program "`return all(v1 < v2 for v1, v2 in zip(tup1, tup2))`" mismatches with the normalized canonical solution "`return all(v1 > v2 for v1, v2 in zip(tup1, tup2))`" at character "<" (highlighted in red). This character maps to the same "<" in the original LLM-generated code "`return all(x < y for x, y in zip(tup1, tup2))`", which is the fifth-generated token by LLM. As a result, the hallucination token index for this example is 5.

As there could be multiple unique normalized canonical solutions per problem, we calculate the hallucination token indices between the LLM-generated program and every unique canonical solution and eventually take the largest hallucination token index.

### 3.4 COLLECTION OF ADDITIONAL SIGNALS FOR HALLUCINATION LOCALIZATION

In addition to the raw generated output, we collect additional signals that could be relevant to hallucination, i.e., per-step log probabilities provided by the LLMs, types of generated tokens, and the error messages of executing the incorrect program.

**Per-step Log Probabilities:** Log probabilities can be obtained during the generation process through LLMs' inference API. The log probs. show the LLMs' confidence level at the corresponding decoding step. We collect the log probs. of the top 100 tokens at each step.

**Token Types:** In programming languages, each token can belong to different categories based on its role in the code, which is analogous to parts of speech in natural language. We categorize tokens of different types to provide code-specific information.

To determine the token types, we parse the code into an abstract syntax tree (AST), where each node has its node type that we use to decide the token type. We classify code tokens, based on AST node types, into the following categories: `Keyword`, `Delimiter`, `Operator`, `Constant`, `Identifier`, and `Type Identifier`. Besides, we also add two additional types: `Space` for the white space tokens and `<EOS>` for the end-of-sequence token (a token that marks the end of generation). Figure 2 shows examples of these token types in Java and Python programs.

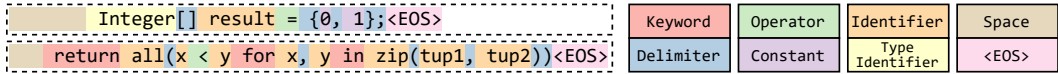

Figure 2: Examples of token types in Java and Python code

**Error Messages:** Execution feedback is crucial for understanding and potentially fixing incorrect code because it usually points to relevant lines where the bug resides. Therefore, we offer the execution feedback of the generated code by running test cases on them. For the CG task, we use EvalPlus (Liu et al., 2023) to run rigorous test cases on the generated code. For the APR task, we use the official evaluation scripts and run the test cases provided by each dataset.

## 4 BENCHMARK ANALYSIS

We present the statistics and analysis of Collu-Bench and show some key findings in this section. Collu-Bench contains 13,234 instances, each with an LLM-generated code, parsed token types, per-step log probs., execution error messages, and the hallucination token index as target (code without hallucination is not included).

### 4.1 ANALYSIS AND FINDINGS

**LLMs are less confident when hallucinating.** Figure 3 shows the probability distributions of correct tokens and hallucinated tokens. (a) shows that for all the LLMs, the hallucinated tokens tend to have a lower probability than the correct tokens. **GPT-4o-mini is much more confident than other LLMs when they are hallucinating.** (b) shows that the code tokens generated for different datasets and tasks still hold the same pattern. Code tokens generated for the HumanEval-Java dataset overall have a higher probability (for both correct and hallucinated ones) than those for other datasets. **Hallucinated tokens generated for CG datasets overall have a lower probability than hallucinated tokens generated for APR datasets.** (c) shows the probability distribution of correct and hallucinated tokens with different types. `Keyword` is the only type that probability distributions of correct and hallucinated tokens overlap the most, suggesting **LLMs are least confident when generating keywords.** And the hallucinated `EOS` tokens have the highest probability, suggesting **LLMs tend to stop generation confidently, even at incorrect places.**

**LLMs are more likely to hallucinate when generating certain types of tokens.** Table 1 shows the error rate of different types of tokens generated by each LLM and for each dataset. Among all the token types, **`Keyword` is the most error-prone type** across all five datasets, and most LLMs (except GPT-4o-mini). Besides, **`Type Identifier` and `Identifier` are also more error-prone for most LLMs** compared to the other types.

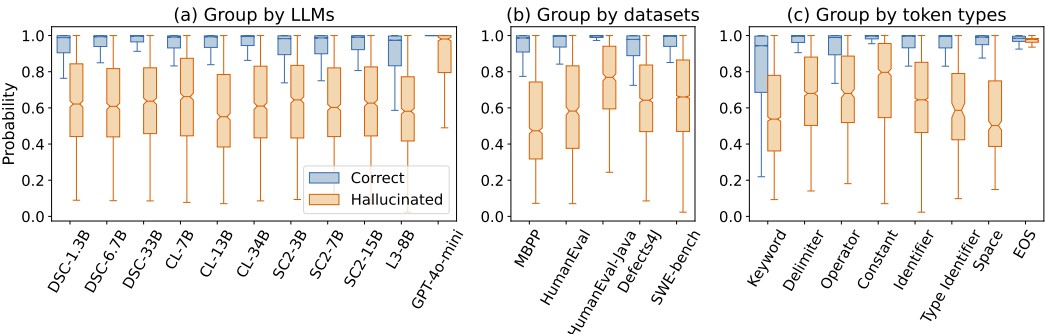

Figure 3: Probability distribution of correct and hallucinated tokens. DSC, CL, SC2, and L3 refer to DeepSeekCoder, CodeLlama, StarCoder2, and Llama3.

When comparing among datasets, Defects4J and SWE-bench data have a much higher hallucination rate in all types of tokens except for `EOS`, which could be due to their complexity. Defects4J is also unique in having a much higher hallucination rate in `Operator`, `Constant`, `Identifier`, and `Type Identifier` tokens.

Table 1: Proportion (%) of hallucinated tokens in each token type generated by each LLM and for each dataset. Token types with $\geq 15\%$, $\geq 10\%$ and $\geq 5\%$ hallucination rate are highlighted.

| | DeepSeekCoder | | | CodeLlama | | | StarCoder2 | | | Llama3 | GPT-4o | | | | | |
| | 1.3B | 6.7B | 33B | 7B | 13B | 34B | 3B | 7B | 15B | 8B | mini | MBPP | HE | HE-Java | D4J | SWE |
|---|---|---|---|---|---|---|---|---|---|---|---|---|---|---|---|---|
| Key. | 14.48 | 11.45 | 10.46 | 15.26 | 14.27 | 12.32 | 15.35 | 13.57 | 11.19 | 14.87 | 8.24 | 6.42 | 5.05 | 4.67 | 22.79 | 22.29 |
| Delim. | 4.36 | 2.77 | 2.23 | 4.30 | 3.80 | 3.17 | 3.99 | 3.29 | 2.84 | 5.72 | 2.38 | 2.68 | 1.82 | 1.93 | 5.91 | 4.72 |
| Op. | 3.62 | 2.75 | 1.91 | 4.52 | 2.68 | 2.87 | 3.70 | 3.77 | 2.71 | 4.11 | 2.08 | 1.69 | 1.39 | 2.35 | 11.11 | 3.60 |
| Const. | 5.84 | 4.13 | 3.15 | 5.38 | 4.51 | 3.74 | 4.97 | 3.66 | 3.61 | 5.44 | 2.37 | 3.25 | 2.51 | 4.39 | 11.90 | 4.32 |
| Id. | 5.66 | 4.38 | 3.72 | 6.13 | 4.58 | 4.35 | 6.04 | 6.38 | 5.00 | 7.70 | 3.78 | 2.52 | 2.35 | 2.46 | 11.92 | 6.97 |
| Type. | 8.33 | 9.09 | 8.88 | 10.91 | 6.58 | 8.49 | 9.42 | 13.59 | 8.96 | 9.33 | 8.81 | 0.00 | 0.00 | 4.27 | 16.06 | 0.00 |
| Sp. | 2.35 | 0.90 | 0.25 | 0.43 | 0.30 | 0.51 | 1.81 | 1.15 | 0.95 | 0.42 | 0.33 | 0.05 | 0.05 | 0.18 | 0.73 | 1.73 |
| EOS | 1.71 | 0.75 | 0.31 | 1.43 | 0.59 | 1.06 | 1.42 | 0.65 | 1.05 | 1.77 | 0.52 | 1.65 | 2.34 | 0.00 | 0.00 | 0.00 |

## 4.2 ERROR RATE

Collu-Bench employs the proposed pipeline (Section 3.1) to automatically identify the first hallucination token as the target. This may not always align perfectly with human developer annotations. To assess the accuracy, we randomly selected 100 samples from Collu-Bench and asked two developers to review the hallucination tokens in the LLM-generated code. The developers disagreed with the identified hallucination tokens in 14 samples and concurred with that of the remaining 86 samples. We then further checked the 14 samples that the developers consider mislabeled and found they were all due to missing a more extensive set of equivalent canonical solutions.

Given the difficulty of identifying code equivalence, it is impossible to exhaustively find and consider all the canonical solutions. Without the proposed solution in Section 3.1, there would only be 57 samples matching the developers' annotation using a simple string match or token match (i.e., 43% error rate). We sample diverse canonical solutions and use program normalization to handle identifier variability, which reduces the error rate of data labeling significantly.

## 5 PRELIMINARY RESULTS OF HALLUCINATION PREDICTION

Collu-Bench can be used to train and evaluate code hallucination localization methods. We formulate the task of code hallucination localization as follows: given a code generated by an LLM, which has been verified to be incorrect by execution test cases, the task is to identify the *first* incorrect token in the generated code. Specifically, given an LLM-generated code $G$, the task is to predict the smallest index $i$ such that $G_i \neq S_i$, where $S$ is the correct solution we expect the LLM to generate.

In this section, we describe our preliminary experiment results on Collu-Bench. We consider the following two task setups:

- **Per-token prediction:** The hallucination prediction model classifies each token as correct or hallucinated, starting from the first token in the LLM-generated code. For an LLM-generated code with hallucination token index $i$, the sample is considered predicted accurately if the prediction model classifies the first $i-1$ tokens as correct and the $i$-th token as hallucinated.
- **Per-sample prediction:** The hallucination prediction model takes all the tokens in the LLM-generated code as input, and selects one from the all as the first hallucination token. A sample with hallucination token index $i$ is considered predicted accurately if the prediction model correctly selects the $i$-th token as the first hallucination token.

For each setup of the hallucination prediction task, we also consider different data split setups:

- **All-in-one**: We apply five-fold cross-validation to split the samples in Collu-Bench into 80% training and 20% test data per fold, and train one prediction model using the training data.
- **One-per-dataset:** Since LLMs may have different patterns in hallucination when generating code for different tasks or datasets, we apply the cross-validation and train one prediction model on data that comes from each dataset independently.
- **One-per-LLM:** Since different LLMs may have diverse patterns in hallucination, we apply the cross-validation and train one prediction model on data from each LLM independently.

## 5.1 PER-TOKEN PREDICTION

We conduct experiments using traditional machine learning (ML) techniques including Support Vector Classifier (SVC), Ada Boost Classifier (AB), Random Forest Classifier (RF), Gradient Boosting Classifier (GB), and Multi-layer Perceptron (MLP). For each token, the considered features include the top 100 probability distribution, the token type (in a one-hot vector), and the token index in the LLM-generated code. Table 2 shows the accuracy of hallucination token index prediction using different models, under the first two data-split settings. We find in general, **RF produces higher accuracy than SVC, AB, GB, and MLP.** When training separate prediction models per dataset, the model (train and test) on SWE-bench produces much higher accuracy than other datasets, and the model on HumanEval produces the worst accuracy, which suggests that **LLMs have different patterns in hallucination when generating code for different task or dataset.**

Table 2: Accuracy (%) of hallucination token index prediction using under "All-in-one" and "One-per-dataset" settings.

| Models | All-in-one | One-per-dataset | | | | |
| --- | --- | --- | --- | --- | --- | --- |
| | | MBPP | HumanEval | HumanEval-Java | Defects4J | SWE-bench |
| Support Vector (SVC) | 32.17 | 26.28 | 7.21 | 29.57 | 30.27 | 37.08 |
| Ada Boost (AB) | 32.02 | 28.55 | 15.77 | 26.21 | 30.98 | 36.40 |
| Random Forest (RF) | **33.09** | **30.61** | 16.73 | **29.69** | **32.27** | 37.62 |
| Gradient Boosting (GB) | 32.74 | 29.87 | 16.73 | 29.07 | 31.69 | **37.86** |
| Multi-layer Perceptron (MLP) | 31.72 | 27.02 | **18.65** | 29.19 | 31.29 | 36.13 |

Table 3: Accuracy (%) under "One-per-LLM" setting. Row names show the LLMs where the training data comes from, and column names show the LLMs where the test data comes from. Accuracy that is $\geq 33\%$, $\geq 31\%$, $\leq 29\%$, and $\leq 27\%$ are highlighted.

| | DSC-1.3B | DSC-6.7B | DSC-33B | CL-7B | CL-13B | CL-34B | SC2-3B | SC2-7B | SC2-15B | L3-8B | GPT-4o-mini |
| --- | --- | --- | --- | --- | --- | --- | --- | --- | --- | --- | --- |
| DSC-1.3B | 30.18 | 31.09 | 29.32 | 28.77 | 31.40 | 30.40 | 30.47 | 27.91 | 29.84 | 24.19 | 8.71 |
| DSC-6.7B | 29.87 | 32.15 | 30.71 | 29.76 | 31.07 | 31.61 | 31.71 | 29.98 | 32.78 | 29.28 | 14.48 |
| DSC-33B | 27.96 | 32.10 | 34.63 | 31.68 | 34.39 | 31.95 | 34.96 | 31.72 | 31.05 | 33.78 | 16.37 |
| CL-7B | 29.21 | 30.33 | 28.58 | 31.03 | 30.23 | 30.14 | 32.25 | 31.25 | 29.23 | 22.07 | 6.09 |
| CL-13B | 27.38 | 28.56 | 33.02 | 29.38 | 32.42 | 30.14 | 30.85 | 28.95 | 29.23 | 32.36 | 5.56 |
| CL-34B | 29.65 | 30.41 | 28.49 | 28.54 | 29.40 | 30.30 | 30.00 | 27.28 | 28.71 | 21.05 | 19.20 |
| SC2-3B | 26.79 | 30.08 | 28.86 | 28.23 | 30.48 | 28.41 | 33.72 | 32.04 | 31.92 | 23.49 | 9.65 |
| SC2-7B | 27.67 | 28.81 | 28.49 | 29.23 | 30.65 | 30.22 | 34.26 | 30.40 | 33.65 | 24.90 | 11.23 |
| SC2-15B | 29.21 | 30.67 | 30.06 | 29.00 | 29.65 | 30.48 | 34.73 | 31.25 | 30.00 | 24.04 | 11.96 |
| L3-8B | 27.38 | 31.93 | 32.65 | 29.99 | 34.88 | 32.04 | 31.47 | 29.58 | 30.44 | 33.62 | 16.16 |
| GPT-4o-mini | 1.24 | 4.89 | 0.56 | 0.92 | 0.75 | 1.47 | 2.87 | 1.11 | 1.82 | 0.47 | 34.21 |

Table 3 shows the accuracy of RF predictors under the "One-per-LLM" settings. (1) **GPT-4o-mini has the most unique pattern in hallucination**, that predictors trained with other LLMs' data predict worse when predicting hallucination in GPT-4o-mini's output, and vice versa. (2) **Predic-**

**tors trained with other LLMs' data in general work worse when predicting hallucination in Llama3-8B's output**, however, predictors trained on Llama3-8B's data generalize successfully to most other LLMs' output except DeepSeekCoder-1.3B and GPT-4o-mini. (3) **Predictor trained with DeepSeekCoder-33B's data generalizes the best and produces higher accuracy on most LLMs' output**, except DeepseekCoder-1.3B and GPT-4o-mini. (4) Surprisingly, **the predictors trained and tested on the data from the same LLMs are not always the most accurate,** e.g., predictor trained with StarCoder2-7B's data are more accurate on predicting StarCoder2-15B's hallucination than predictor trained with StarCoder2-15B's data (33.65% versus 30.00%).

## 5.2 PER-SAMPLE PREDICTION

For per-sample prediction, we conduct experiments using the same three settings. The predictors take a list of tokens in the LLM-generated code, the feature of each token includes the top 100 probabilities and token type in a one-hot vector. The predictors encode the token list using CNN (Lecun et al., 1998), RNN, LSTM (Hochreiter & Schmidhuber, 1997) or GRU (Cho et al., 2014)), or Transformer (Vaswani et al., 2017) layers to produce hidden states for each token. The hidden states of the token list are fed to a pointer network (Vinyals et al., 2017; Hossain et al., 2024b) to select the first hallucination token from the list.

Table 4 shows the accuracy of hallucination token index prediction using the above neural network (NN) models. **LSTM shows the highest accuracy under the "All-in-one" setting**, and **under the "One-per-dataset" setting, CNN produces the highest accuracy** on data collected from most datasets (HumanEval-Java and SWE-bench). Besides, compared with per-token prediction, **LSTM under per-sample prediction achieves similar accuracy to RF under the "All-in-one" setting (33.09% versus 33.15%).** On data collected from each dataset, **ML approaches with per-token prediction are much more accurate than neural networks with the per-sample prediction on MBPP, but are less accurate on HumanEval-Java.**

Table 4: Accuracy (%) of hallucination token index prediction using Collu-Bench under "All-in-one" and "One-per-dataset" settings.

| Models | All-in-one | MBPP | HumanEval | One-per-dataset HumanEval-Java | Defects4J | SWE-bench |
|---|---|---|---|---|---|---|
| CNN | 32.30 | 23.42 | 17.90 | **42.86** | 29.04 | **38.38** |
| GRU | 32.85 | **24.05** | 17.48 | 40.91 | 28.07 | 36.97 |
| LSTM | **33.15** | 21.52 | 17.90 | 36.36 | **31.19** | 37.98 |
| Transformer | 23.03 | 20.89 | **20.09** | 35.71 | 26.12 | 27.14 |

Table 5: Accuracy (%) under "One-per-LLM" setting. Row names show the LLMs where the training data comes from, and column names show the LLMs where the test data comes from. Accuracy that is $\geq 35\%$, $\geq 33\%$, $\geq 31\%$, $\leq 29\%$, and $\leq 27\%$ are highlighted.

| | DSC-1.3B | DSC-6.7B | DSC-33B | CL-7B | CL-13B | CL-34B | SC2-3B | SC2-7B | SC2-15B | L3-8B | GPT-4o-mini |
|---|---|---|---|---|---|---|---|---|---|---|---|
| DSC-1.3B | 36.46 | 32.80 | 33.78 | 35.36 | 31.34 | 33.89 | 28.68 | 23.81 | 26.20 | 29.46 | 0.00 |
| DSC-6.7B | 35.38 | 32.80 | 30.67 | 37.26 | 31.95 | 31.38 | 28.29 | 24.21 | 30.57 | 31.78 | 0.00 |
| DSC-33B | 34.30 | 34.40 | 31.56 | 36.89 | 32.78 | 33.89 | 31.01 | 25.79 | 28.82 | 32.17 | 0.10 |
| CL-7B | 35.38 | 31.60 | 32.00 | 35.74 | 31.12 | 30.96 | 28.29 | 21.83 | 25.76 | 30.62 | 0.00 |
| CL-13B | 38.63 | 30.40 | 30.67 | 38.02 | 34.02 | 30.23 | 28.97 | 29.26 | 29.26 | 31.78 | 0.00 |
| CL-34B | 35.74 | 31.60 | 31.56 | 37.64 | 30.71 | 31.38 | 29.46 | 26.59 | 30.13 | 31.40 | 0.00 |
| SC2-3B | 34.30 | 32.00 | 29.78 | 34.22 | 33.20 | 32.22 | 31.40 | 29.76 | 36.24 | 32.56 | 0.49 |
| SC2-7B | 35.74 | 32.00 | 32.89 | 34.22 | 32.78 | 31.80 | 29.46 | 31.35 | 34.50 | 28.68 | 0.49 |
| SC2-15B | 35.38 | 34.40 | 31.56 | 38.02 | 33.61 | 34.31 | 31.78 | 35.32 | 34.93 | 33.33 | 0.00 |
| L3-8B | 33.94 | 34.00 | 31.56 | 34.98 | 31.12 | 32.22 | 30.62 | 29.76 | 31.44 | 28.68 | 0.00 |
| GPT-4o-mini | 1.44 | 0.40 | 1.33 | 0.00 | 0.41 | 0.00 | 0.00 | 0.40 | 0.00 | 0.78 | 35.61 |

Table 5 shows the accuracy of the LSTM predictors under the "One-per-LLM" setting. Except for the same conclusion that "GPT-4o-mini" has the most different pattern from other LLMs, NNs under "per-sample prediction" draw dissimilar findings than ML approaches. (1) Overall, **NNs show higher upper bound than ML approaches under the "One-per-LLM" setting**, with many predictors producing accuracy higher than 35%. (2) Hallucination of DeepSeekCoder-1.3B, which is hard to predict in the per-token manner, can be predict more accurate in the per-sample manner. This suggests the per-token and per-sample prediction approaches could complement each other.

## 6 LIMITATION

One limitation is the errors in the target hallucination token index provided in Collu-Bench, which is determined by an automated pipeline and thus is non-perfect. Compared with simple string matching or token matching, we sample diverse canonical solutions and apply program normalization to handle the equivalency and identifier variability of code to increase the accuracy of the hallucination token index in Collu-Bench significantly. It is non-trivial to find an automated solution to determine the hallucination in code perfectly, which remains to be explored.

Another limitation is the range of select LLMs and datasets to build Collu-Bench. There exist lots of different LLMs and code generation or program repair datasets, we select the set of state-of-the-art, widely-used LLMs (including DeepSeekCoder series, CodeLlama series, StarCoder2 series, Llama3 series, and GPT-4o-mini), and dataset. Overall, Collu-Bench's 13,234 data samples come from 11 LLMs' output on five datasets. Studying the hallucination of more LLMs and datasets can be an interesting future work.

## 7 CONCLUSION

This work presents Collu-Bench, a challenging benchmark for code hallucination localization. Collu-Bench includes 13,234 hallucination instances generated by 11 diverse LLMs on two important code tasks, offering a comprehensive evaluation of hallucination localization across multiple models. Collu-Bench also provides additional information such as per-step log probs. produced by LLMs, types of generated tokens, and execution feedback as useful signals for predicting code hallucinations. Through extensive experiments using traditional machine learning techniques and neural network models as hallucination predictors, we provide an in-depth study of hallucination localization using Collu-Bench. The preliminary results reveal that traditional ML methods and neural networks can only achieve an accuracy of up to 33.15%, highlighting the complexity of this task, and underscoring the need for further research in improving the trustworthiness and reliability of LLMs in code-related applications.

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

## A  APPENDIX

### A.1  DETAILS OF PROGRAM NORMALIZATION

Table 6 lists the AST nodes in Python and Java languages that refer to code containing user-defined identifiers. The underscored identifiers are those we collected in each example.

On average, after sampling diverse canonical solutions and normalizing program, we collected 82.01, 50.01, 5.54, 1.31, and 1.53 unique normalized canonical solutions per problem in HumanEval, MBPP, HumanEval-Java, Defects4J, SWE-Bench.

Table 6: AST nodes that contain user-defined identifiers (underscored) in Python and Java programs.

| Python AST Nodes | Examples | Java AST Nodes | Examples |
|---|---|---|---|
| assignment | `x = 1` | variable declarator | `int x = 0;` |
| for statement | `for x in nums:` | enhanced for statement | `for (Integer i : nums)` |
| for in clause | `[x**2 for x in nums]` | lambda expression | `nums.sort((a, b) -> b.compareTo(a));` |
| with statement | `with open(...) as fp:` | method declaration | `int add(int x, int y)` |
| except clause | `except Exception as e:` | constructor declaration | `Point(int x, int y)` |
| lambda | `lambda x: x**2` | | |
| function definition | `def add(x, y):` | | |

System Prompt
```
You are an exceptionally intelligent coding assistant that consistently delivers accurate and
reliable responses to user instructions.
```

Five-shot Examples
```
### Task Start ###
Complete the function `similar_elements` below.

```python
def similar_elements(test_tup1, test_tup2):
    """ Write a function to find the similar elements from the given two tuple lists.
    >>> similar_elements((3, 4, 5, 6), (5, 7, 4, 10))
    (4, 5)
    >>> similar_elements((1, 2, 3, 4),(5, 4, 3, 7))
    (3, 4)
    >>> similar_elements((11, 12, 14, 13),(17, 15, 14, 13))
    (13, 14)
    """
    res = tuple(set(test_tup1) & set(test_tup2))
    return res
```

### Task Start ###

...
```

Test Sample
```
### Task Start ###
Complete the function `square_nums` below.

```python
def square_nums(nums):
    """ Write a function to find squares of individual elements in a list using lambda function.
    >>> square_nums([1, 2, 3, 4, 5, 6, 7, 8, 9, 10])
    [1, 4, 9, 16, 25, 36, 49, 64, 81, 100]
    >>> square_nums([10, 20, 30])
    [100, 400, 900]
    """
```

LLM's Output
```
    return list(map(lambda x: x**2, nums))
```

Figure 4: Few-shot prompt we used to collect LLMs' outputs for code generation tasks

## A.2 FEW-SHOT PROMPTING DESIGN

Figures 4 and 5 show the few-shot prompts we used during the collection of LLMs' outputs. For the code generation task, we follow the prompt format in HumanEval that provides the task description and example inputs and outputs as a doc-string inside the function signature.

For the automated program repair task, we provide the task description which is important to understand the intention of the function. The original buggy code is enclosed by <bug> and </bug> to separate from the surrounding context. The LLMs are only required to generate the corresponding fixed code to replace the buggy code.

In the prompt, all the source code is also enclosed by "```" followed by the programming language, which is commonly used in Markdown files. Such a design enables us to distinguish the end of code generation in time using "```" as the stop word and prevent LLMs from generating further explanations or comments.

**System Prompt**

```
You are an exceptionally intelligent coding assistant that consistently delivers accurate and
reliable responses to user instructions.

You will be provided with a text description outlining a problem, the function that is intended
to solve the problem yet contains a bug, with the erroneous code highlighted between <bug> and
</bug> tags. Your task is to analyze the entire function and the buggy code, then generate the
corrected version of the buggy code.

The generated fixed code will directly replace the buggy code within the function. Please ensure
that the syntax is correct and that no additional code is produced beyond the fixed code, as this
could lead to syntax errors when the fixed code is inserted back into the function.
```

**Five-shot Examples**

```
### Task Start ###
* Problem Description
Check if in the given list of numbers, are any two numbers closer to each other than
given threshold.

Examples:
has_close_elements([1.0, 2.0, 3.0], 0.5) returns false
has_close_elements([1.0, 2.8, 3.0, 4.0, 5.0, 2.0], 0.3) returns true

* Function
```java
public class ROLLING_MAX {
    public static List<Integer> rolling_max(List<Integer> numbers) {
        List<Integer> result = new ArrayList<Integer>();
        Integer running_max = null;

        for (Integer n : numbers) {
<bug>
                double distance = numbers.get(i) - numbers.get(j);
</bug>

                if (distance < threshold)
                    return true;
            }
        }
        return false;
    }
}
```

* Buggy Code
```java
                double distance = numbers.get(i) - numbers.get(j);
```

* Fixed Code
```java
                double distance = Math.abs(numbers.get(i) - numbers.get(j));
```

### Task Start ###
```

```
...

### Task Start ###
* Problem Description
Return list of all prefixes from shortest to longest of the input string

Examples:
all_prefixes("abc") returns ["a", "ab", "abc"]

* Function
public class ALL_PREFIXES {
    public static List<String> all_prefixes(String string) {
        List<String> result = new ArrayList<String>();

        for (int i = 0; i < string.length(); i += 1) {
<bug>
            result.add(string.substring(i + 1));
</bug>
        }
        return result;
    }
}

* Buggy Code
```java
            result.add(string.substring(i + 1));
```

* Fixed Code
```java
```

**Test Sample**
```

**LLM's Output**

```
            result.add(string.substring(0, i + 1));
```

Figure 5: Few-shot prompt we used to collect LLMs' outputs for program repair tasks.

## A.3 DETAILS OF SELECTED LLMs

Table 7 shows the details of our selected LLMs, including their release date, pre-training data size, and the number of parameters. CodeLlama is developed by Meta AI, training the Llama2 models (which have already been trained on 2T natural language tokens) using an additional 700B code tokens. DeepSeekCoder uses the same architecture as Llama, yet it trained from scratch using 2T tokens, 13% of which is natural language text and 87% is code tokens. StarCoder2 is developed by the BigCode project, as an evolution of the original StarCoder (Li et al., 2023c) model, optimized for multi-language support and fine-tuned for a variety of programming tasks. CodeLlama, DeepSeek-Coder, and StarCoder2 are specialized in source code, performing well on various code tasks such as code generation, code infilling, and supporting multiple programming languages.

Llama3 is the latest generation of Meta's Llama models pre-trained with significantly more data (15T tokens), although it is a general LLM not specialized for source code, it shows strong capability in both natural language and code.

GPT-4o-mini is an optimized version of GPT-4, developed by OpenAI, to support strong reasoning on both natural language text and code, and also keep high efficiency with smaller. It is one of the strongest commercial LLM. The training data and process of GPT-4o-mini are unknown.

| Models | Release Date | Pre-training Size | Parameters |
|---|---|---|---|
| CodeLlama | Aug. 24, 2023 | 2T NL tokens and 700B code tokens | 7B
13B
34B |
| DeepSeekCoder | Jan. 26, 2024 | 2T tokens (13% NL and 87% code) | 1.3B
6.7B
33B |
| StarCoder2 | Feb. 28, 2024 | 3.3T NL and code tokens
3.7T NL and code tokens
4.3T NL and code tokens | 3B
7B
15B |
| Llama 3 | April 18, 2024 | 15T NL and code tokens | 8B |
| GPT-4o-mini | July 18, 2024 | - | - |

Table 7: The release dates, pre-training data, and number of parameters of selected LLMs.

## A.4 ADDITIONAL STATISTICS OF COLLU-BENCH

Table 8 lists the detailed number of instances collected from each LLM and each dataset in Collu-Bench. The data collected from each LLM is relatively balanced, while the data collected from each dataset is imbalance, with SWE-bench contributing the most data.

Table 9 presents the proportion of each token type in the code generated by each LLM, and the proportion of each token type in the code generated for each dataset. All LLMs consistently generate the most tokens for `Identifier` (32.98 − 36.95%). All DeepSeekCoder and CodeLlama models generate similar proportions of tokens for `Delimiter` and `Space` (∼ 20%). The rest models share a similar pattern in that they generate around 19.48 − 23.04% tokens for `Delimiter` and 12.16 − 14.44% tokens for `Space` and `Constant`.

Generated code for all the datasets contains most tokens for `Identifier`, with simpler datasets (MBPP, HumanEval, HumanEval-Java) having 25.77 − 28.03% and more complex datasets (Defects4J and SWE-bench) having 32.43 − 37.38%. For CG datasets, `Space` is the second most types and `Delimiter` is the third most. By contrast, for APR datasets, the second and third most common types are `Delimiter` and `Space`.

## A.5 PARAMETER TUNING OF HALLUCINATION PREDICTION MODELS

In Seciotns 5.1 and 5.2, we train traditional machine learning models and neural networks to predict code hallucination using Collu-Bench as the dataset.

For per-token prediction, since the number of correct tokens is much more than the number of hallucination tokens, we down-sample the correct tokens to prevent the predictor from overfitting to

Table 8: Number of instances in Collu-Bench that collected from each LLM and dataset.

| Models | DeepSeekCoder | | | CodeLlama | | | StarCoder2 | | | Llama3 | GPT-4o | Total |
|---|---|---|---|---|---|---|---|---|---|---|---|---|
| | 1.3B | 6.7B | 33B | 7B | 13B | 34B | 3B | 7B | 15B | 8B | mini | |
| MBPP | 200 | 148 | 126 | 219 | 190 | 172 | 184 | 177 | 159 | 184 | 140 | 1899 |
| HumanEval | 114 | 83 | 70 | 116 | 102 | 101 | 110 | 106 | 92 | 115 | 32 | 1041 |
| HumanEval-Java | 97 | 78 | 51 | 89 | 70 | 70 | 85 | 85 | 55 | 87 | 37 | 806 |
| Defects4J | 220 | 202 | 200 | 204 | 203 | 197 | 206 | 206 | 202 | 213 | 191 | 2254 |
| SWE-bench | 735 | 676 | 679 | 679 | 637 | 618 | 687 | 687 | 645 | 675 | 553 | 7234 |
| Total | 1366 | 1187 | 1081 | 1307 | 1204 | 1158 | 1290 | 1261 | 1153 | 1274 | 953 | 13,234 |

Table 9: Proportion (%) of each token type generated by each LLM and for each dataset. The first , second , and third most types by each LLM or for each dataset are highlighted. Key., Delim., Op., Const., Id., Type., and Sp. refer to Keywords, Delimiter, Operator, Constant, Identifier, Type Identifier, and Space. HE, D4J, and SWE refer to HumanEval, Defects4J, and SWE-bench.

| | DeepSeekCoder | | | CodeLlama | | | StarCoder2 | | | Llama3 | GPT-4o | MBPP | HE | HE-Java | D4J | SWE |
|---|---|---|---|---|---|---|---|---|---|---|---|---|---|---|---|---|
| | 1.3B | 6.7B | 33B | 7B | 13B | 34B | 3B | 7B | 15B | 8B | mini | | | | | |
| Key. | 5.86 | 5.55 | 5.37 | 5.29 | 5.34 | 5.52 | 6.54 | 6.38 | 6.62 | 6.53 | 7.70 | 8.17 | 9.54 | 4.23 | 5.53 | 5.42 |
| Delim. | 20.73 | 20.86 | 20.17 | 20.18 | 19.35 | 20.58 | 22.98 | 23.55 | 23.04 | 20.39 | 19.48 | 20.77 | 19.23 | 25.37 | 24.39 | 20.38 |
| Op. | 5.24 | 5.45 | 5.40 | 4.78 | 4.56 | 4.88 | 5.53 | 5.49 | 5.41 | 5.47 | 6.19 | 7.84 | 7.98 | 11.32 | 6.98 | 4.01 |
| Const. | 10.91 | 12.19 | 13.29 | 12.39 | 12.66 | 11.06 | 13.08 | 13.68 | 13.15 | 12.16 | 13.44 | 10.53 | 11.59 | 7.98 | 7.69 | 13.85 |
| Id. | 35.06 | 34.99 | 34.82 | 33.86 | 32.98 | 34.23 | 35.83 | 35.58 | 36.95 | 34.96 | 35.49 | 28.03 | 25.77 | 27.16 | 32.43 | 37.38 |
| Type. | 0.54 | 0.50 | 0.51 | 0.54 | 0.36 | 0.53 | 0.48 | 0.55 | 0.39 | 0.50 | 0.67 | 0.00 | 0.00 | 2.64 | 3.39 | 0.00 |
| Sp. | 20.25 | 18.99 | 19.12 | 21.75 | 23.66 | 21.78 | 13.82 | 13.20 | 12.81 | 13.82 | 14.44 | 22.20 | 24.24 | 18.58 | 17.53 | 17.24 |
| EOS | 1.41 | 1.48 | 1.31 | 1.20 | 1.09 | 1.42 | 1.72 | 1.55 | 1.63 | 1.68 | 2.58 | 2.46 | 1.64 | 2.72 | 2.06 | 1.17 |

correct tokens. We tune the ratio of correct and hallucination tokens in the range of 1: 1 to 10: 1, and eventually use 3: 1 in the final experiments due to its best performance. For other hyper-parameters of SVC, RF, AB, GB, and MLP, we use the default provided in scikit-learn [2].

For per-sample prediction, we tune the hyper-parameters of each architecture accordingly (e.g., the number of layers, hidden dimensions, etc.). The final CNN models have four stacked convolution layers and a hidden dimension of 512. Both the LSTM and GRU models have two bidirectional layers and a hidden dimension of 512. The transformer models have four layers, with a hidden dimension being 256 and a feed-forward dimension of 1024. The attention layers in the transformers have eight attention heads. Each model is trained with a batch size of 32 for 10 epochs, using Adam as the optimizer to update the weights.

---

[2] https://scikit-learn.org/stable/

