# OpenReview forum: "Collu-Bench: A Benchmark for Predicting LLM Hallucinations in Code"
_ICLR.cc/2025/Conference — ICLR 2025 Conference Withdrawn Submission_

### Official Review · Reviewer_GHSr · 2024-10-28

**Soundness:** 2
**Presentation:** 2
**Contribution:** 2
**Rating:** 3
**Confidence:** 4

**Summary:**

The paper constructs a benchmark containing hallucinated code generated by different LLMs. It also annotates the positions of hallucination tokens, aiming to identify where the model starts exhibiting hallucination behavior. The authors analyze from the perspective of model confidence and the types of hallucinated tokens, discovering corresponding patterns—for instance, models generally have lower confidence when dealing with hallucinated tokens. Additionally, they use some basic machine learning and deep learning models to identify model hallucinations and evaluate these predicters in different settings.

**Strengths:**

The paper is easy to follow.

From the perspective of model confidence, the paper identifies distinct patterns between hallucinated tokens and correctly generated tokens.

**Weaknesses:**

1. I believe that "hallucination in code" is fundamentally an ill-defined term, and it is inherently challenging to define. Specifically, in this work, it seems that hallucinated code and incorrect/buggy code are treated as entirely equivalent. Therefore, I think using this term without a rigorous definition is neither precise nor reliable.

2. The finding that models exhibit low confidence on hallucinated tokens is very interesting. However, relying solely on token confidence to achieve high identification accuracy is insufficient. Currently, the performance of per-token prediction and per-sample prediction is quite similar, which indicates that the model heavily depends on the confidence feature for identification. However, I believe that this task should be analyzed more from the semantic perspective of the code, which might achieve higher accuracy. For instance, a naive approach, such as having the model review its own generated code, might yield decent identification accuracy.


3.This task does not seem fundamentally different from bug localization or program review. The objective in all cases is to identify parts that do not meet the code generation requirements. Program review, in particular, is even more challenging as it involves not only identifying but also correcting these parts.

4.Even though the authors considered diverse canonical solutions, I believe that using text-based comparisons for data annotation remains imprecise, as there is no guarantee that the range of canonical solutions covers all possible solutions adequately.

**Questions:**

1. Before comparing the hallucinated code generated by the model with canonical solutions, do you use methods such as unit tests or program analysis to determine whether the code does not meet the intended generation?

2. Did you remove comments when processing model-generated data, as many models, such as GPT-4, may include annotations for the generated statements?

3. refer to weakness 3, what are the differences between this benchmark and tasks like program review or bug localization?

---

### Official Review · Reviewer_47va · 2024-11-04

**Soundness:** 2
**Presentation:** 3
**Contribution:** 2
**Rating:** 5
**Confidence:** 3

**Summary:**

This study introduces Collu-Bench, a benchmark specifically designed to identify and analyze hallucinations in code generated by large language models (LLMs), addressing gaps in current research on code hallucinations. Collu-Bench includes 13,234 instances from five datasets produced by 11 different LLMs, focusing on two key tasks: code generation (CG) and automated program repair (APR). It provides detailed features such as per-step log probabilities, token types, and execution feedback for fine-grained analysis and prediction. Experiments using traditional machine learning and neural network models achieve a maximum accuracy of 33.15%, underscoring the challenge of this task. Findings reveal that LLMs show lower confidence in hallucinated outputs and are more prone to hallucinations with specific token types, highlighting the need to improve LLM reliability and accuracy in code generation.

**Strengths:**

Collu-Bench includes a comprehensive set of 13,234 instances across diverse LLM models and coding tasks.

Provides valuable, fine-grained data such as log probabilities, token types, and execution feedback to support hallucination analysis.

Experiments reveal key patterns, like low confidence during hallucinations and higher hallucination rates for specific tokens.

**Weaknesses:**

Achieved accuracy limits immediate applicability in practical settings.

Excludes state-of-the-art models, potentially reducing relevance to newer LLM architectures.

Focuses only on code generation and repair, missing other critical coding applications affected by hallucinations.

Identifies patterns but lacks actionable approaches to reduce hallucinations in practice.

**Questions:**

How effective is the automated sampling process in capturing a comprehensive set of canonical solutions, especially for more complex tasks in Defects4J and SWE-bench datasets?

What are the limitations of the program normalization technique in accurately detecting hallucinations? Are there instances where the normalization process might incorrectly standardize genuinely distinct solutions?

In cases where the generated code subtly deviates from the canonical solutions, how does Collu-Bench ensure that the hallucination token is accurately identified without oversimplifying or introducing false positives?

What criteria were used to select the five specific datasets, and how might additional datasets impact Collu-Bench’s robustness and versatility?

This paper includes 11 LLMs of various sizes and types. What is the reasoning behind selecting these specific models, and how might the inclusion of more recent or specialized models impact the benchmark’s findings?

Why do certain token types, like Keywords and Identifiers, appear more susceptible to hallucinations? Could this be influenced by the specific training data or architecture of the LLMs?

The analysis highlights different hallucination patterns across datasets, such as Defects4J showing a high hallucination rate for Operators and Identifiers. What underlying factors in these datasets contribute to these distinct hallucination profiles?

How does the per-token prediction approach compare with a per-example prediction regarding interpretability and practical application? Are there scenarios where one approach is more advantageous?

Traditional ML models like Random Forest perform better in specific setups, while neural networks excel in others. What characteristics of hallucination prediction tasks make certain model types more suitable, and could a hybrid model improve results?

The highest accuracy achieved was around 33.15%. What are the main barriers to achieving higher accuracy, and are there known model improvements or alternative feature sets that could be integrated to boost predictive performance?

---

### Official Review · Reviewer_JqwD · 2024-11-04

**Soundness:** 3
**Presentation:** 2
**Contribution:** 2
**Rating:** 5
**Confidence:** 3

**Summary:**

This paper successfully introduces Collu-Bench, a challenging benchmark for code hallucination localization. It includes 13,234 hallucination instances generated by 11 diverse LLMs across five datasets, offering a comprehensive evaluation of hallucination localization across multiple models. Furthermore, Collu-Bench provides additional information such as per-step log probabilities produced by LLMs, types of generated tokens, and execution feedback, which are useful signals for predicting code hallucinations. Through extensive experiments using traditional machine learning techniques and neural network models as hallucination predictors, the paper provides an in-depth study of hallucination localization using Collu-Bench. Preliminary results indicate that traditional ML methods and neural networks can only achieve an accuracy of up to 33.15%, highlighting the complexity of this task and emphasizing the need for further research to improve the trustworthiness and reliability of LLMs in code-related applications.

**Strengths:**

1. The paper clearly defines the problem of code hallucination in LLMs and provides a comprehensive benchmark for research in this area. 2. The inclusion of diverse LLMs and datasets is a significant contribution to the field.
3. The paper presents a well-structured approach to collecting and analyzing code hallucination instances. The automated pipeline for handling program equivalency and identifier variability is innovative and adds value to the benchmark.
4. The experiments conducted using traditional machine learning techniques and neural networks are thorough and provide valuable insights into the patterns of code hallucination. The findings highlight the challenges and potential areas for future research.

**Weaknesses:**

1. Code models generate hallucinatory code, what kind of code can be referred to as hallucinatory code? The definitions of hallucinatory code and hallucinatory tokens in the text are inaccurate. In the abstract section,  the authors mention "content that sounds plausible but is actually incorrect", this definition is too vague. In the construction of Collu-Bench, the authors consider samples that fail to pass test cases as hallucinatory code and the first token that differs from the standard solution as the hallucinatory token. This is clearly not accurate enough. Failing to pass test cases indicates that the code is incorrect, but it does not necessarily mean it is hallucinatory code.

2. While less attention has been given to hallucinations in source code as mentioned in the abstract, there are still several works that address this issue. The paper needs to compare the Collu-Bench dataset with other efforts, such as CodeMirage and CoderEval, to highlight their differences.
- CodeMirage: Hallucinations in Code Generated by Large Language Models. https://arxiv.org/abs/2408.08333
- CoderEval: A Benchmark of Pragmatic Code Generation with Generative Pre-trained Models. https://arxiv.org/abs/2302.00288

3. Hallucinatory code should be deceptive code that appears reasonable to humans but is actually incorrect. How can we ensure that the code sampled from LLMs that fails to pass test cases is also seemingly reasonable to humans and likely to be misused?

4. The purpose of the dataset is to reduce the likelihood of LLMs generating hallucinatory code. However, the dataset is primarily used to enhance the model's ability to predict hallucinatory code and hallucinatory tokens. Enhancing the model's predictive capabilities for hallucinatory code and tokens does not necessarily reduce the probability of LLMs generating hallucinatory code.

5. Does the normalization process of the code in this paper potentially destroy or lose the semantics of the original code?

6. In the process of constructing the dataset, it is taken for granted that code that fails to pass test cases is considered hallucinatory code. In reality, such code is not equivalent to hallucinatory code. The dataset constructed in this way contains both "hallucinatory code" and "code with obvious errors that do not cause hallucinations." If "code with obvious errors that do not cause hallucinations" is not excluded, then the dataset itself has issues, and all subsequent results lack a solid foundation.

7. The extent to which LLMs produce hallucinatory code in the dataset construction lacks explanation. Why are some LLMs more prone to generating hallucinatory code, while others are not as likely to produce such code?

8. The article mentions and briefly compares CodeHalu and HalluCode, both of which classify and define code hallucinations. However, the concept of hallucinatory code in this paper is vague. The authors should also provide a detailed definition of the concept of hallucinatory code and categorize them.

9. The results of various experimental models on the Collu-Bench dataset lack detailed explanations. Why do some methods perform poorly/well, and what are the reasons for their poor/good performance?

10. The Collu-Bench dataset currently covers only Java and Python languages. It would be beneficial to construct a dataset that includes more mainstream programming languages, such as C, C++, and Go.

11. Consider conducting a more overall human evaluation of the dataset's quality and the accuracy of annotations.

12. The dataset relies on LLMs for annotation, but LLMs are not fully reliable, this may lead to incorrect token locations. How to identify and correct errors in the dataset?

13. Despite the reduction, the error rate remains relatively high, with 14 out of 100 randomly sampled instances flagged as questionable. How can the error rate be further lowered?

14. The paper could benefit from a more detailed discussion of the implications of the findings and how they relate to existing work in the field.

**Questions:**

1. How can we ensure that the code sampled from LLMs that fails to pass test cases is also seemingly reasonable to humans and likely to be misused?
2. Does the normalization process of the code in this paper potentially destroy or lose the semantics of the original code?
3. Why are some LLMs more prone to generating hallucinatory code, while others are not as likely to produce such code?
4. Why do some methods perform poorly/well, and what are the reasons for their poor/good performance?
5. How to identify and correct errors in the dataset?

---

### Official Review · Reviewer_SD48 · 2024-11-05

**Soundness:** 2
**Presentation:** 2
**Contribution:** 1
**Rating:** 3
**Confidence:** 4

**Summary:**

The paper presents Collu-Bench, a benchmark for detecting code hallucinations in outputs from large language models. With over 13,000 instances from 11 models, it helps assess hallucination localization using various data points. It highlights the challenge and need for improved LLM reliability in coding.

**Strengths:**

- The paper provides a dataset with rich information to analyze hallucination in coding tasks
- The authors reveal patterns of code hallucinations across data sources and LLMs

**Weaknesses:**

- The method for ground truth hallucination localization is overly simplistic and may not apply to complex cases, despite the method proposed in section 3.1 (see Questions)
- The finding of "LLMs are less confident when hallucinating" is not novel and has been widely used for detecting hallucinations, e.g. [1], [2], [3], to name a few. However, I appreciate the authors' experiments studying finer-grained hallucination positions in coding tasks. The authors should emphasize more on their new findings specifically on this domain.
- The localization methods only take the probability distribution of top-100 tokens into account, without considering the semantic meanings of the tokens, nor the execution feedbacks.
- More hallucination detection baselines should be discussed and compared.
- Lack of discussion of the proposed "code hallucination" vs bug localization.

[1] Xiao, Yijun, and William Yang Wang. "On Hallucination and Predictive Uncertainty in Conditional Language Generation." Proceedings of the 16th Conference of the European Chapter of the Association for Computational Linguistics: Main Volume. 2021.

[2] Guerreiro, Nuno M., Elena Voita, and André FT Martins. "Looking for a Needle in a Haystack: A Comprehensive Study of Hallucinations in Neural Machine Translation." Proceedings of the 17th Conference of the European Chapter of the Association for Computational Linguistics. 2023.

[3] Zhang, Tianhang, et al. "Enhancing Uncertainty-Based Hallucination Detection with Stronger Focus." Proceedings of the 2023 Conference on Empirical Methods in Natural Language Processing. 2023.

**Questions:**

- How do you define hallucination on coding tasks? How does it compare to the bug localization task?

- Discussions about hallucination localization in dataset creation:
  - The per-token hallucination localization method in section 3.3 still looks weak to me after canonical solution sampling. The proposed methods addresses the problem of "identifier variability", but how to tackle the problem of semantically identical problems? For example, how do you detect the hallucination location if the ground truth is `return all(v1 > v2 for v1, v2 in zip(tup1, tup2))` and the generation is
    ```python
    for v1, v2 in zip(tup1, tup2):
        if not v1 < v2:
            return False
    return True
    ```
  An error rate of 14% is reported in section 4.2. How does this affect the usability of the dataset? Is it possible to provide a clean subset of the dataset to train localizers to figure out the impact of wrong annotations?


  - In section 3.3,
  >  As there could be multiple unique normalized canonical solutions per problem, we calculate the hallucination token indices between the LLM-generated program and every unique canonical solution and eventually take the largest hallucination token index.

  What is the reason and how accurate is the design of taking the largest index? Moreover, how do you handle multiple hallucinations in the code? Will keeping only one hallucination index cause false negatives training detectors?

- Table 1 shows a major source of hallucinations is *keyword*. However, is it related to the process of program normalization?

---

### Official Review · Reviewer_e4aw · 2024-11-08

**Soundness:** 3
**Presentation:** 3
**Contribution:** 3
**Rating:** 5
**Confidence:** 5

**Summary:**

The authors introduce Collu-Bench, a benchmark designed to evaluate code hallucinations in LLMs. This benchmark includes 13,234 instances of code hallucinations from 11 different LLMs across five datasets, covering both code generation and automated program repair tasks. Collu-Bench’s innovation lies in its automated process that combines program equivalence and identifier variation to locate hallucinated tokens accurately. The benchmark provides detailed signals, including the log probability at each step, token types, and execution feedback. The authors conduct preliminary experiments using traditional machine learning and neural network methods to predict hallucinations, with prediction accuracy ranging from 22.03% to 33.15%. Overall, this benchmark aims to advance the understanding, prediction, and mitigation of hallucinations in automated code generation and program repair tasks.

**Strengths:**

1. Collu-Bench differs from previous benchmarks by focusing on finer-grained code hallucinations, providing a new benchmark that includes richer features such as log probabilities and execution feedback. It aims to deepen understanding and predict where hallucinations occur.

2. The authors write the paper clearly, emphasizing the importance of the problem. The structure of each section is well-organized, making it easy to understand the motivation, methodology, experimental setup, and conclusions of Collu-Bench.

3. The authors execute their experiments effectively, from benchmark construction to analysis and results. They offer detailed descriptions of the findings, complemented by visualizations of experimental results, which enhance the persuasiveness of the conclusions.

**Weaknesses:**

1. The authors provide an introduction in Section 3 on how Collu-Bench is constructed and how they generate the Ground Truth. However, I am concerned about the accuracy and quality of the Ground Truth generation method. Despite performing a manual review, the authors achieve only an 86% accuracy rate, which introduces potential bias during evaluation. Moreover, the sample size for manual verification (100 samples) is relatively small compared to the dataset’s scale. How do the authors address the issue of low Ground Truth quality?

2. The detection of hallucinations relies on comparing the generated code with a "standard" solution, which may not cover all possible correct solutions, potentially leading to inaccurate hallucination detection. How do the authors address this issue to ensure more accurate hallucination detection?

3. In Sections 5.1 and 5.2, the authors merely describe the experimental results without providing detailed analysis. Could they offer more specific insights into why these experimental results occur? For example, why does GPT-4o-mini exhibit the most unique hallucination patterns? Why does the predictor trained on Llama3-8B data generalize well to content generated by most other LLMs? And why do Transformer models perform with relatively low accuracy on Collu-Bench?

**Questions:**

1. Could the authors provide specific case studies? Do they examine whether certain types of programming tasks or problem structures are more likely to trigger hallucinations? Providing a more detailed error analysis would be helpful, especially in cases where hallucinations are misidentified or overlooked. Are there specific features or patterns that lead to these errors?

3. The authors present a large evaluation dataset, which in practice may make it challenging for researchers with limited computational resources to replicate the results. For instance, the authors themselves do not use all 2,294 entries in SWE-Bench. Do the authors have any specific measures to address this issue?

**Details Of Ethics Concerns:**

None.

---

### Note · Authors · 2024-11-27

I have read and agree with the venue's withdrawal policy on behalf of myself and my co-authors.